# Alzheimer’s Disease and Inflammaging

**DOI:** 10.3390/brainsci12091237

**Published:** 2022-09-13

**Authors:** Anna Mikhailovna Kosyreva, Alexandra Vladislavovna Sentyabreva, Ivan Sergeevich Tsvetkov, Olga Vasilievna Makarova

**Affiliations:** Avtsyn Science Research Institute of Human Morphology of FSBI “Petrovsky National Research Centre of Surgery”, 117418 Moscow, Russia

**Keywords:** inflammation, neurodegeneration, aging, atherosclerosis, metabolic syndrome, depression, Alzheimer’s disease

## Abstract

Alzheimer’s disease is one of the most common age-related neurodegenerative disorders. The main theory of Alzheimer’s disease progress is the amyloid-β cascade hypothesis. However, the initial mechanisms of insoluble forms of amyloid-β formation and hyperphosphorylated tau protein in neurons remain unclear. One of the factors, which might play a key role in senile plaques and tau fibrils generation due to Alzheimer’s disease, is inflammaging, i.e., systemic chronic low-grade age-related inflammation. The activation of the proinflammatory cell phenotype is observed during aging, which might be one of the pivotal mechanisms for the development of chronic inflammatory diseases, e.g., atherosclerosis, metabolic syndrome, type 2 diabetes mellitus, and Alzheimer’s disease. This review discusses the role of the inflammatory processes in developing neurodegeneration, activated during physiological aging and due to various diseases such as atherosclerosis, obesity, type 2 diabetes mellitus, and depressive disorders.

## 1. Introduction

Aging is a complex, dynamic, multistage, and inevitable biological process that leads to a gradual decrease in the adaptive capacity of the body. It is characterized by the development of so-called age-related pathology and an increased probability of death. At the biological level, aging results from various molecular and cellular damage that accumulates over time [1]. The number of elderly and senile people is steadily growing from year to year; therefore, among them, the proportion of patients with age-associated diseases, which are socially significant and increase the healthcare burden, is increasing. These diseases include type 2 diabetes mellitus (T2DM), atherosclerosis, cardiovascular, and neurodegenerative diseases, including the most common type of dementia, which is Alzheimer’s disease (AD). The mechanism of its development is still unclear; however, there is growing data on the relationship of AD with chronic inflammatory diseases, as well as age-associated inflammation, called inflammaging.

## 2. Aging and Inflammation

The average age of the population of European countries is the highest in the world, and it is increasing. According to WHO forecasts, from 2020 to 2030, the share of people over 60 within the world population will increase by 34%. By 2050 the number of people aged 80 and over will triple, reaching 426 million [1].

Despite the existence of many theories of aging, none of them fully reveals the fundamental mechanisms underlying this complicated process [2]. Theories on aging can be conditionally divided into two main groups: evolutionary and accidental cell damage. The first group includes theories according to which aging is a programmed process that an organism has acquired during evolution. For example, there are theories of antagonistic pleiotropy, adaptive-regulatory, disposable soma, telomeric, immunological, etc. The second combines theories that consider the accumulation of accidental damage as the main cause of aging. Examples of such theories are intoxication, mitochondrial, epigenetic, apoptosis, accumulation of mutations, etc. [3,4].

In the process of aging, 3 types of changes are observed. The primary one is associated with disorders in the genome, such as a high frequency of mutations, shortening of telomeres (Hayflick limit), epigenetic changes, such as methylation and acetylation of DNA sections, and, as a result, a violation of protein homeostasis in cells. The second type of age-associated change is secondary, realized due to the occurrence of primary ones. This group includes mitochondrial dysfunction and cellular senescence [5]. The third type of changes are integrative, such as depletion of the pool of stem cells and disturbances in intercellular interactions [6]. It is customary to distinguish three types of aging—natural or physiological, premature or pathological, and delayed.

Regardless of the initial mechanisms, the aging of the body is accompanied by systemic chronic low-grade inflammation, which in foreign literature is referred to as inflammaging, i.e., inflammation associated with age [7]. Inflammaging was described in all mammalian species, including laboratory rodents, rhesus monkeys [8], and humans [9]. Some authors associate physiological and pathological aging with the hyperproduction of proinflammatory cytokines and inflammatory mediators produced by innate immune cells [10]. Inflammaging is supposed to reduce life expectancy, especially if combined with age-associated diseases such as T2DM, cardiovascular, oncological and neurodegenerative diseases, including AD [11]. In these diseases, concomitant obesity is often observed, against the background of which the severity of inflaming increases [12].

Cell aging is associated with the impossibility of exiting the G1 or G2/M phase of the cell cycle. This process is accompanied by abrupt changes in cell size, shape, and vacuolization of the cytoplasm, as well as functional properties, including an impaired rate of decline, rearrangement of the nucleus and chromatin, and resistance to the apoptosis signal [13]. However, these cells remain functionally active and produce a certain proinflammatory secretion, known is SASP (senescence-associated secretory phenotype), which is characteristic of aging cells. SASPs include proinflammatory cytokines (IL-1α, IL-1β, IL-6, IL-8, TNFα), chemokines (CCL2, CCL5, CCL20), growth factors (TGF, EGF, bFGF, HGF, VEGF), metalloproteinases (MMP-1, -3, -10, -12, -13, -14), extracellular matrix components (fibronectin, collagen, laminin), aging-associated beta-galactosidase (SA-β-Gal), etc. The composition and intensity of SASP expression depend on the cell type, the triggers that led to its activation, and the time elapsed from its onset. With age, the level of expression of the proinflammatory secretome SASP progressively increases, which leads to the accumulation of products of cellular metabolism. These include free radicals, extracellular ATP, nuclear non-histone protein HMGB1 (high-mobility group protein B1), uric acid, products of the impaired metabolism of phospholipids, and cell membrane proteins, which are ceramides, cardiolipin, lipofuscin, and amyloid-β (Aβ), proteins with irregular spatial organization, such as α-synuclein and tau protein, fragments of mitochondrial and nuclear DNA. All this leads to the activation of proinflammatory reactions and the development of low-grade chronic inflammation [14]. An imbalance in the production of pro- and anti-inflammatory cytokines due to cellular aging leads to the formation of a “senile” phenotype and the development of age-associated diseases [15]. Along with proinflammatory reactions in the aging body, the activation of anti-inflammatory mechanisms is observed. According to the literature, compared with young and middle-aged people, older people have an increased level of both proinflammatory cytokines such as IL-1, IL-2, IL-6, IL-8, IL-12, IL-15, IL-17, IL-18, IL-22, IL-23, TNF-α, and IFN-γ, and anti-inflammatory ones like IL-1Ra, IL-4, IL-10, IL-37, and TGF-β1. The imbalance is a probable cause of the development of inflammatory diseases [16].

It was shown that the expression of NF-κB, a nuclear factor that activates the production of proinflammatory cytokines, increases with age in humans and laboratory animals [17]. Compared with mature laboratory rodents, the skin, liver, kidneys, cerebellum, cardiomyocytes, and gastric mucosa NF-κB binding to DNA regions is more stable, which determines its prolonged effect on the expression of target genes [18]. Skin fibroblasts of elderly people and patients with a genetic form of premature aging, known as childhood progeria (Hutchinson-Gilford syndrome), are characterized by a high level of NF-κB activation [19].

With age, the efficiency of eliminating senescent cells by immunocompetent cells decreases, partly due to involutive changes in the immune system. As a result, the number and duration of functioning of SASP-secreting cells increases, ultimately leading to the aging of tissues and the body altogether, and also increases the risk of developing age-related diseases, and AD in particular [20].

Thus, cellular aging is a risk factor for developing age-related diseases, including neurodegenerative ones. There are many theories of aging; however, despite the differences in the initial mechanisms underlying these theories, ultimately, cellular aging is characterized by the activation of the proinflammatory phenotype of SASP-producing cells. A shift in the balance of SASP towards proinflammatory cytokines may be one of the key mechanisms for the development of chronic inflammatory diseases associated with age, such as atherosclerosis, metabolic syndrome, T2DM, and AD.

## 3. Hypothesis of the Development of Alzheimer’s Disease

Alzheimer’s disease is one of the most common neurodegenerative diseases, clinically characterized by progressive deterioration of cognitive functions, memory loss, and irreversible changes in human behavior [21]. AD is classified as an age-related disease since the prevalence of AD increases with age. So, among 65–75-year-olds, about 10% of people suffer from this disease, and among people over 80 years old, the index reaches 32%. According to the WHO, the number of patients with AD is steadily increasing, and the total number of people with dementia is predicted to be about 82 million people in 2030 and 152 million by 2050 [22].

AD was first described in 1906 by Alois Alzheimer. He followed up on a 51-year-old patient with cognitive impairment, disorientation, hallucinations, and other behavioral changes over a 5-year period. He revealed diffuse atrophy of the cortex and “partial changes in cortical cells” in the morphological study of the brain after her death [23]. Despite the fact that AD was described more than a hundred years ago, its clinical diagnosis was developed only in 1984, and its criteria revised in 2011 and 2018 [24]. However, the differential diagnosis of AD with other neurodegenerative diseases is still complicated.

### 3.1. Genetic Hypothesis of AD Development

One of the genetic hypotheses of the development of AD and the accumulation of Aβ is the presence of three mutations. It is the presenelin-1 gene (Presenilin-1, PS1) and presenelin-2 (Presenilin-2, PS2), which encode the components of γ-secretase, and the gene for the precursor of Aβ, known as APP (Amyloid-β precursor protein) [21]. However, less than 5% of all AD cases are associated with genetic disorders and the presence of these mutations in patients. These cases are classified as a familial form, or early onset AD (EOAD) when they manifest before 65 years of age [25]. The most common form of AD is the senile one, characterized by late onset (LOAD), in which symptoms appear in people over 65. Some genetic disorders are also observed in the pathogenesis of this form of AD, but mostly it affects the so-called risk genes, which mainly regulate innate immunity reactions and the functioning of microglial cells [26]. These include the following: BIN1 (Bridging Integrator 1), a protein responsible for the regulation of intracellular vesicle sorting, modulation of β-secretase transport (BACE-1; b-site APP cleavage enzyme 1), and Aβ production [27]; clusterin, or apolipoprotein J (CLU), a protein of the small chaperone family involved in folding, i.e., the spatial orientation of the amyloid-β; triggering receptor expressed on myeloid cells 2 (TREM2), which activates the production of proinflammatory cytokines; CD33, or Siglec-3, sialic acid binding Ig-like lectin 3, that activates the expression of the membrane receptor TREM2 by microglial cells; type 1 complement receptor—CR1, or C3b/C4b; an enzyme of the phospholipase C family, phospholipase C gamma 2 (PLCG2); ABI3, which is a protein that activates cell migration, etc. [28].

The most compelling evidence for polymorphisms of inflammatory mediators that increase the risk of AD is associated with the IL1 complex. Chromosome 2q14-21 contains a cluster of IL1-related genes, including IL1A, IL1B, and the IL1 receptor antagonist protein (IL-1RA). Grimaldi et al. found an association between the polymorphisms of the IL-1A gene and AD onset. So the IL-1A T/T genotype doubles the risk for AD, with carriers of this genotype showing an onset of AD 9 years earlier than IL-1A C/C carriers. [29]. Polymorphisms of the IL-1 genotypes may confer risk for Alzheimer’s disease through IL-1 overexpression and IL-1–driven neurodegenerative cascades [30].

In addition to genetic polymorphisms, early and late forms of AD have several differences in microglial activation and white matter atrophy. According to Taipa et al., EOAD has higher microglial activation in subiculum, entorhinal, and temporal subcortical white matter than LOAD [31]. Greater white matter atrophy has been reported in EOAD than in LOAD, probably reflecting a more aggressive form of the disease [32]. However, in another study, it was shown that biomarkers of inflammation (TNFα, IL-6, IP-10 (CXCL10), IL-10) were not significantly different between patients with early and late onset of AD and were only associated with age [33].

### 3.2. Hypothesis of the Amyloid Cascade in AD Development

One of the dominant theories of AD development is the “hypothesis of the amyloid cascade” [34], which links the pathogenesis of AD with the accumulation of various forms of Aβ in the brain. Amyloid Precursor Protein (APP) is a type-1 transmembrane protein synthesized by many cell types in the body. In the CNS, APP folds sequentially in two local pathways, non-amyloid and amyloid. In the first case, APP is cleaved by α-secretase with the formation of an extracellular secreted APP associated with the membrane of the α-C-terminal fragment. Further destruction by γ-secretase leads to the formation of the APP intracellular domain (AICD). Aβ is formed from APP via the amyloid pathway, in which APP is cleaved into two domains by β-secretase. Then, γ-secretase cleaves it into a membrane-bound fragment—AICD and an extracellular product—Aβ, consisting of 37–42 amino acids.

Aβ monomers aggregate and form oligomers, protofibrils, and amyloid fibrils, which are large and difficult to degrade fibers that can form amyloid plaques [35]. Unlike fibrils, protofibrils and oligomers are soluble. Therefore they can penetrate the blood-brain barrier (BBB) and spread through the nervous tissue of the brain due to binding to receptors for advanced glycation end products (RAGE) [36], apolipoprotein E (LRP1) [37], glycoprotein 330 [38], and P-glycoprotein [39]. The mechanism of a switch from non-amyloid to amyloid APP cleavage remains unclear. However, it is supposed that overproduction of soluble forms of Aβ or deterioration of its clearance by microglia leads to its aggregation and the formation of insoluble fibrils, which then form amyloid plaques.

According to the literature, soluble Aβ and amyloid plaques have toxic properties [40]. They have a negative effect on neighboring cells, activating hyperphosphorylation of the tau protein, which leads to the formation of neurofibrillary tangles (NFT) [41]. Normally, the tau protein is involved in maintaining the transport network of the neuron cytoskeleton by binding to tubulin and forming a stable form of microtubules [42]. If an excess amount of Aβ is present in the extracellular space, the tau protein is hyperphosphorylated, which leads to its oligomerization, as a result of which microtubules become unstable and dissociate into fragments. Ultimately, large fragments of tau filaments form NFT, which is insoluble, strong fibrils. Their accumulation in the cytoplasm of the soma and processes of neurons leads to the loss of synaptic contacts, disruption of impulse conduction, and cell death [39]. Tau degradation and phosphorylation is regulated by the soluble form of Aβ and different kind of kinases, including glycogen synthase kinase 3 (GSK3β) and cyclin-dependent kinase 5.

The formation of amyloid plaques and NFTs accompanies the activation of microglial production of proinflammatory cytokines and the release of damage-associated molecular patterns (DAMPs). They also exhibit excitotoxicity, which leads to an increase in ROS production and the development of oxidative stress [43]. These processes eventually end with the death of neurons, mainly cholinergic ones and glia [44]. The loss of cholinergic neurons is especially important in the nucleus basalis of Meynert. Approximately 90% of nucleus basalis neurons are cholinergic, a main source of cholinergic innervation of the cerebral cortex [45]. This structure is vulnerable to the deteriorating impact of NFTs on neurons [46]. Its loss eventually leads to the destruction of axons traveling from the nucleus basalis of Meynert to the cortical surface connecting different cortex zones, and the disturbance of these white matter pathways worsens cognitive impairment in AD patients [47]. Noteworthy, TNFα was identified as a regulator of differentiation of neurons in the nucleus basalis of Meynert under physiological conditions. However, its excessive production, which also depends on epigenetic factors, has a negative influence on neurogenesis, potentiating cell death of the nucleus basalis of Meynert cholinergic neurons in particular [48].

### 3.3. Prions Hypothesis of AD Development

However, there is an opinion that all of the above processes of formation of Aβ and NFTs can develop independently of the formation of amyloid plaques [2]. Several authors consider that Aβ and tau protein, as well as α-synuclein, which accumulate in Lewy bodies localized in the substantia nigra in patients with Parkinson’s disease, can exhibit the properties of prions. They can initiate spontaneous reassembly of proteins and stochastic refolding, resulting in the formation of a stable misfolded spatial organization [49], and then trigger misfolding in other unaltered endogenous proteins in an avalanche-like way. This process is called templating [50], when the modified amyloid becomes a matrix and activates the misfolding of soluble amyloid or other proteins. This process leads to resistance to external influences, including resistance to proteases, heating to 98 °C, denaturation with 2% sodium lauryl sulfate solution, etc. [51]. In a normally functioning cell, chaperones, also known as heat shock proteins (HSPs), are responsible for assembling and maintaining the correct conformation of proteins. Chaperones can prevent improper protein aggregation and the transport of proteins with an altered spatial organization. Depending on the molecular weight, HSPs are subdivided into small HSPs (co-chaperones, small HSP, sHSP), whose molecular weight does not exceed 40 kDa, and larger HSPs, represented by the HSP70, HSP80, HSP90, and HSP100 families [52].

There are two ways of Aβ disaggregation via chaperones. The ATP-dependent way implements when chaperones of the HSP70, HSP80, HSP90, and HSP100 families hydrolyze fibrils in the amyloid plaque. The ATP-independent one is mediated by sHSP [52]. According to the literature, Hsp70 promotes the correct folding of the tau protein and amyloid, thereby preventing the formation of insoluble protein aggregates. It also activates the degradation and dephosphorylation of the tau protein of an abnormal conformation [53,54]. On an AD model in old transgenic 5xFAD mice with increased expression of APP and PSEN1, it was found that intranasal administration of Hsp70 reduced the number of dying neurons with signs of karyolysis, nuclear pyknosis, cytolysis, and vacuolization of the cytoplasm in the hippocampus and temporal cortex compared with mice in the control group [54]. In turn, Hsp90 inhibits amyloid aggregation, preventing its transition to an insoluble form, and activates the clearance of insoluble forms of amyloid by microglia. Small chaperones, such as SHPB1, SHPB5, SHPB6, and SHPB8, are found within amyloid plaques and have been demonstrated to inhibit the oligomerization of amyloid monomers, thus preventing the formation of new fibrils. The effect of SHPB1 and SHPB on the tau protein was also described: sHSP binding to oligomers of the tau protein slows down its aggregation into fibrils but does not prevent it [55]. It was shown that proteins with abnormal spatial organization accumulate during aging in most cells, including neurons. A compensatory increase in the expression of chaperones was detected, but their content and activity also decreased with age, which can contribute to the development of age-associated diseases, including AD [56].

### 3.4. Oestrogen Hypothesis of AD Development

Noteworthy, the female sex is a great risk factor for LOAD [57], and susceptibility to aging-related neurodegenerative diseases is greater in women than in men in general [58]. It is probably related to the «estrogen hypothesis». In the brain, estrogen and oestradiol regulate glucose metabolism, glycolysis, oxidative phosphorylation, and ATP generation in neurons, promote mitochondrial bioenergetics in the brain and calcium homeostasis, reduce ROS production, and protect cells from apoptosis. It also provides anti-inflammatory regulating, including on microglial cells [59]. Downregulation in the estrogen level during peri- and post-menopause leads to disturbances in all these processes hitherto requiring a higher concentration of this hormone. The dysregulation of glucose metabolism in the brain and subsequent accumulation of DAMPs, together with activation of the immune system, including microglia, eventually induce chronic low-grade inflammation and an increase in its level [59].

### 3.5. Inflammation and AD Development

Some authors consider inflammation as the initial mechanism of neurodegeneration in AD [60,61]. In patients with AD and in experimental models of this disease, an increase in the production of cytokines in the brain was found, as well as activation of glial cells, in particular microglia and astrocytes [62]. Also, an increase in the number of glial cells was observed directly in the locus of senile plaques in patients with AD [63,64]. This indicates an important role of the inflammatory response and glia activation in the process of neurodegeneration. Il-1a, IL-1b, IL-6, TNFa, a2-macroglobulin, and a1-antichymotrypsin are upregulated in the tissue of patients with AD and prominently associated with AD lesions [65].

In recent years, more data on the activation of the innate immune system have been revealed in the pathogenesis and progression of neurodegenerative diseases, including AD [66,67]. It was established that the genes encoding immune cell receptors, such as complement receptor type 1—CR1, CLU, Siglec-3 or CD33, and TREM2, are associated with AD and aging [68]. The development of AD is also determined by genes that modulate immune responses both in the CNS and in the body in general. They are non-receptor tyrosine kinase SYK, granulin, which regulates the growth and proliferation of nerve cells (GRN), fructose transporter solute carrier family 2 member 5 (SLC2A5), TNFα-dependent signaling pathway suppressor, pyrin domain containing 1 (PYDC1), hexosaminidase A lysosomal enzyme beta subunit (HEXB), and adapter protein involved in B-lymphocyte differentiation, B cell linker (BLNK) [69].

Neuroinflammation can also contribute to the production of Aβ through an IFITM3 (interferon-induced transmembrane protein 3)-γ-secretase complex [70]. It has been reported that aging, the biggest risk factor for AD, induces type I IFNs that modulate brain function [71]. Therefore, aging-induced neuroinflammation could lead to an increase in the level of IFITM3, which potentiates γ-secretase activity for Aβ production in humans [70]. γ-Secretase consists of four obligatory subunits, presenilin (PS), nicastrin (Nct), Aph-1, and Pen-2 [72]. IFITM3 binds to PS1 in the proximity of the active site and upregulates γ-secretase for Aβ production in a subpopulation of LOAD patients and, therefore, could be a potential risk factor for AD [70].

The brain is regarded as an immunologically privileged organ. It is protected from systemic effects by tight junctions of the endothelial cells of the blood-brain and hemato-liquor barriers, as well as the brain-cerebrospinal fluid barrier (through the arachnoid and ependyma) [73]. However, many studies have revealed that acute bacterial or viral diseases develop systemic inflammatory responses that affect the functional state of the brain through the activation of neural and humoral pathways by inflammatory mediators [55]. One of the mechanisms for the initiation and development of AD is systemic inflammation. As summed up in [74], there is an association between systemic inflammation induced by bacterial and viral infections and AD progression in humans and laboratory rodents. It was shown that intravenous administration of LPS at a dose of 1 mg/kg once to C57BL/6J mice increased the level of Aβ in the hippocampus and led to cognitive impairment [55]. Low-grade systemic inflammation induced by LPS also causes microgliosis and neuronal dysfunction in a second-generation mouse model (*App^NL-G-F^*) of AD [75]. Some studies demonstrate an association between periodontal bacteria or *H. pylori* infection and AD in humans [76,77,78]. AD patients show elevated antibodies against periodontal bacteria [79,80] and higher anti-*H. pylori* IgG titers in their blood and brain [76]. Some studies associated virus infections with cognitive disturbances, dementia, and AD. So it was shown that HSV infection is associated with a higher risk for dementia and was isolated from the brains of AD patients [81,82,83,84].

### 3.6. Autoimmune Hypothesis of AD Development

In recent years, researchers have paid attention to the autoimmune component of AD development. Autoimmunity may arise from dysregulated self-tolerance mechanisms or pathogen mimicry of AD-related proteins [85]. Antibodies generated in response to infection may inadvertently have an affinity to human proteins with homologous regions, leading to auto immunization. Pathological autoantibodies have a neurotoxic effect by inducing tissue damage through autoimmune reactions, promoting microglia-mediated inflammation, and leading to neurodegeneration and AD [85]. In the post-mortem brain in AD, the number of immunoglobulin-positive neurons increases compared with controls [86]. Immunoglobulin-positive neurons can promote Aβ plaque formation and have been associated with microglia-induced neuronal death by the classical antibody-dependent complement pathway [86,87,88,89,90,91]. On the other hand, autoantibodies may play a protective role by promoting amyloid clearance and inhibiting toxic aggregation [85,92,93].

### 3.7. Hypoxia and AD Development

In recent years, hypoxia has been considered one of the risk factors for the development of AD [94]. The production of proteins that regulate oxidative stress, inflammation, apoptosis, mitochondrial metabolism, and synaptic transmission is activated with a lack of oxygen in cells, which leads to neuron death [95,96]. In response to hypoxia, the hypoxia-inducible factor 1 (HIF1) is stabilized, which activates the transcription of several genes involved in cell adaptation to oxygen deficiency [97]. Prolonged exposure to tissue hypoxia can initiate many neurodegenerative diseases, including AD [94]. Hypoxia-induced accumulation of Aβ due to changes in the expression level of enzymes involved in the destruction of APP and disruption of calcium homeostasis in glial and nerve cells is considered one of the mechanisms for the development of AD [98]. Li et al. [99], in a study on APPswe + PS1A246E transgenic mice, showed that the formation of amyloid plaques is accelerated due to the activation of β- and γ-secretase and APP cleavage along the amyloid pathway under hypoxic conditions. Jakubauskienė et al. [100] found that hypoxia leads to changes in the 3R/4R ratio in tau protein mRNA in vitro, but not in APP, both in neurons and glial cells, which confirms the importance of pre-mRNA splicing in the development of neurodegenerative diseases.

HIF-1, the content of which increases in the cell during hypoxia, can also perform a neuroprotective function. This transcription factor activates the expression of proteins that increase capillary permeability and improve microcirculation, including erythropoietin, glucose transporter 1,3, and the vascular endothelial growth factor (VEGF) [101]. Erythropoietin can inhibit Aβ-initiated neuronal apoptosis, while glucose transporter 1,3 activates glucose transport into nerve cells.

One of the risk factors for the development of neurodegenerative diseases in the postnatal period can be fetal hypoxia. It was shown that hypoxia in the antenatal period leads to an increase in the number of microglial and astroglial cells in the cerebral cortex of mice, as well as to cognitive disorders, but does not affect the accumulation of Aβ in the cortex and the hippocampus [102].

Thus, there are various hypotheses concerning the development of AD, and today it is clear that the etiology and pathogenesis of this disease multiply. In addition to the multiple mechanisms in the pathogenesis of AD, various factors play an equally important role, e.g., lifestyle, chronic stress, chronic diseases, including cardiovascular ones and atherosclerosis, obesity, T2DM, depression, and so on [103].

## 4. Alzheimer’s Disease and Chronic Inflammatory Diseases

Over the past decades, many researchers have noted that AD is more likely to develop in people with chronic diseases, including atherosclerosis, metabolic syndrome, T2DM and depressive disorders. All these diseases are characterized by systemic manifestations of chronic inflammation, similar to inflammaging, which is supposed to play a triggering role in the formation and accumulation of Aβ in AD.

### 4.1. Atherosclerosis and Alzheimer’s Disease

Atherosclerosis is a chronic inflammatory disease of the arteries of the elastic and muscular-elastic type, which occurs as a result of a violation of lipid and protein metabolism. It is accompanied by the deposition of cholesterol and some fractions of lipoproteins in the arterial wall with the formation of atherosclerotic plaques. Around the deposits of cholesterol and lipoproteins, inflammatory infiltration is observed with the growth of connective tissue and the development of atheromatosis and calcification. All this leads to deformation and stenosis of the lumen of the arteries. Atherosclerosis has risk factors similar to AD: age, dyslipidemia, arterial hypertension, T2DM, smoking, and the presence of the apolipoprotein 4 (APOE4) allele.

According to Roher et al. [104], stenosis of the coronary arteries of atherosclerotic origin is observed more often in patients with AD than in the comparison group of corresponding age. At the same time, the severity of atherosclerosis of the vessels of the circle of Willis correlated with the severity of pathological changes in brain tissues, such as the total number of amyloid plaques, the number of NFTs, and atrophy index of white matter assessed by MRI [104], as well as with cognitive impairment [105]. In another work by Roher et al., it was found that AD is more often accompanied by cerebral atherosclerosis compared to non-Alzheimer type dementia, which the authors attribute to impaired Aβ utilization, due to the development of a hypoxic state [104]. According to Urbanova et al. [105], the thickness of the intima-muscular membrane complex in the arteries in patients with AD is significantly higher than in elderly people without AD. However, in other studies, the relationship between the severity of carotid artery stenosis and cognitive impairment in patients with both AD and without any neurodegenerative diseases was not detected [6]. This may be due to the heterogeneity of study groups of patients with atherosclerosis by age and gender.

The pathogenesis of both atherosclerosis and AD is closely related to the activation of inflammatory and immune responses [16]. Thus, in atherosclerosis, an increase in the expression of the inflammatory marker CRP is observed in 90% of atheromatous plaques in response to an increase in the production of proinflammatory cytokines IL-1 and IL-6 [106]. In AD, an increase in the content of CRP in the blood serum or cerebrospinal fluid was not found. However, at the early stages of the development of the disease, a 20-fold increase in the expression of CRP mRNA was detected in brain tissues, particularly in the hippocampus [107].

According to Guo et al. [108], the development of both AD and atherosclerosis is associated with the activation of the NLPR3 inflammasome, a cytosolic multiprotein complex that initiates pyroptosis, or programmed necrotic cell death, via a caspase-1-dependent signaling pathway. An experimental AD model in APP/PS1 transgenic mice deficient in NLRP3 and caspase-1 expression showed that their neurodegenerative changes were less pronounced. These mice were characterized by reduced brain production of Aβ, proinflammatory cytokines, and less cognitive impairment as assessed in the Morris water maze, open field, and novel object recognition test. At the same time, an increase was observed in the expression of arginase-1 and IL-4, markers of microglial polarization according to the M2 anti-inflammatory phenotype [109].

Activation of the NLRP3 inflammasome also occurs in atherosclerosis. It was shown in vitro that deposits of cholesterol and low-density lipoproteins formed during atherosclerosis in the vessel wall interact with toll-like receptors type 4 (TLR4) and activate NLRP3, which leads to an increase in the production of proinflammatory cytokines IL-1β and IL-18 in vitro [110] and in vivo [108,111].

It should be noted that NLRP3-dependent activation of IL-18 production is involved not only in the pathogenesis of AD and atherosclerosis. It was revealed that in Wistar rats with metabolic syndrome, intravenous administration of IL-18 resulted in the activation of monocyte recruitment into the aortic wall [112]. In addition, the authors showed that an increase in the expression of IL-18 in the aorta in rats with metabolic syndrome leads to an increase in insulin resistance (IR) due to the activation of the proinflammatory NF-κB signaling pathway [112]. According to the literature, the activation of NF-κB-dependent production of pro-inflammatory cytokines correlates with the formation of amyloid plaques and the progression of clinical manifestations of AD [113].

### 4.2. Obesity and Alzheimer’s Disease

Obesity is a chronic disease characterized by excessive accumulation of adipose tissue in the body, as well as metabolic disorders. Obesity was shown to increase the expression of many proinflammatory markers, including CRP [114], plasminogen activator-1 inhibitor [115], IL-6 and TNFα [116], monocyte chemokine CCL2, or MCP-1 [117], and acute phase protein—serum amyloid [118]. The development of obesity is often associated with cardiovascular disease and T2DM [119].

Recently, the number of studies confirming the association of obesity/metabolic syndrome with neurodegenerative changes in the brain has increased in the literature. According to clinical studies, patients with metabolic syndrome and/or obesity are almost twice as likely to develop AD [120]. According to Anstey et al. [121], with obesity, the risk of developing AD, vascular, and other forms of dementia increases by 35, 33, and 26%, respectively. The molecular mechanisms underlying the effect of adipose tissue hypertrophy on neurodegeneration processes are related to APP production in adipose tissue in the metabolic syndrome increases, as well as levels of Aβ and cholesterol in the blood serum. It can eventually lead to increased Aβ transport in the brain and impaired functioning of neurons [122,123].

In the metabolic syndrome accompanying obesity, the inflammatory reaction in adipose tissue is aseptic, chronic, and low-grade, which distinguishes it from the reaction of the organism to pathogen presence. According to the literature, in obesity, the accumulation of neutral lipids in adipocytes with subsequent infiltration by macrophages leads to hypoxia and oxidative stress. It initiates the development of inflammatory reactions due to the activation of the JNK/activator protein 1 (AP1) and IKK/NF-κB signaling pathways [124]. As shown in experimental models and humans, JNK [125] and IKK [126] phosphorylate AP1 (c-Jun/Fos) and NF-κB lead to the production of proinflammatory cytokines and a decrease in insulin sensitivity with the development of IR [127]. Inflammatory responses in adipose tissue and IR affect the metabolism of all organs, including the liver, muscles, and pancreas [124,128]. Willette et al. [129] found that the severity of IR correlated with the prevalence of Aβ deposits in the prefrontal and temporal cortex. A high level of proinflammatory cytokines, including IL-1β and IL-6, observed in the blood of obese patients, can initiate damage to the BBB and affect its permeability. This might lead to an increase in the production of proinflammatory cytokines by glial cells and vascular endothelium and the development of neurodegenerative diseases as a result [130].

It should be noted that cholesterol, the content of which is associated with obesity and metabolic syndrome, has a high affinity for APP and Aβ [129]. In addition, β- and γ-secretases are found in lipid raft zones, areas of the plasma membrane rich in cholesterol. Therefore, its activity influences these enzymes’ activity and, as a result, amyloid and non-amyloid pathways of Aβ formation [130,131,132].

### 4.3. Type 2 Diabetes Mellitus and Alzheimer’s Disease

Along with obesity, many researchers have identified a correlation between the development of AD in patients with T2DM. According to Arvanitakis et al. [133], patients with T2DM have a 65% higher risk of developing AD than those without one. A recent meta-analysis of 17 longitudinal cohort studies amounting to about 1.7 million people concluded that DM increases the risk of developing AD with a relative risk of approximately 1.5 [134]. Li et al. [135] showed that, in animals fed a high-fat diet, obesity and the development of T2DM were combined with a decrease in the proliferation and differentiation of neuro-progenitor cells (NPCs) in the hypothalamus compared with mice fed a standard diet. The authors attribute the decrease in the number of NPCs in mice with T2DM to their apoptotic death due to the high level of expression of the proinflammatory cytokines TNFα and IL-1β [135].

The development of T2DM is associated with long-term persistent hyperglycemia and tissue hypoxia, in which end products of glycation accumulate, such as pentosidine and glyceraldehyde-derived pyridinium (GLAP) [136]. They increase the expression of BACE1, which leads to the activation of APP metabolism through the amyloid pathway and the accumulation of Aβ [137]. According to Choi et al. [138], an age-dependent increase in the expression level of the Ager gene, encoding RAGE receptors, was observed in the hippocampus and prefrontal cortex of 3xTg-AD transgenic mice, expressing human APP, Aβ, and tau protein. RAGEs are also pattern-recognition receptors (PRRs). A high level of RAGE expression in old 3xTg-AD mice correlated with an increase in the expression of the marker of microglial activation—Iba1 [138]. Another study showed that transgenic mutant mice with a high expression of RAGE and APP exhibited disturbances in spatial memory, a decrease in the excitatory postsynaptic potential in synapses in the CA1 field of the hippocampus, and a decrease in the number of synaptophysin-positive and acetylcholinesterase-positive axons [139]. HSP70 and the nuclear non-histone protein HMGB1 can activate RAGE, which, as in the case of TLR4 activation, leads to NF-κB-dependent production of proinflammatory cytokines, including TNF-α, and an increase in ROS production [140].

In patients with diabetes, there are pathological changes in blood vessels with the destruction of BBB, and cerebral endothelial cell dysfunction, reducing cerebral blood flow and thus eventually leading to brain parenchymal damage [141]. According to Liang et al. [141], diabetes is the risk factor for AD and sporadic cerebral small vessel disease (CSVD). It is known that with age, large artery stiffness increases, which could also lead to blood vessel dysfunction, resulting in neuropathology and cognitive impairment [142]. In a large cohort of women diagnosed with breast cancer at age l, with up to 26 years of follow-up from 1991 to 2016, it was found that cerebrovascular disease (CVD), stroke, and diabetes are associated with a significantly higher risk of AD [143]. It is possible that another additional mechanism in the development of AD is a violation of the cerebral microvasculature; however, this factor is not decisive, and often cerebrovascular diseases are not accompanied by the development of AD.

According to Farris et al. [144], the insulin-degrading enzyme (IDE) regulates the destruction of Aβ and APP. In hyperinsulinemia, the IDE reserve is consumed only for the breakdown of insulin. Therefore, the rate of metabolism of APP and Aβ decreases, and their concentration increases. It was shown there is a decrease in Aβ degradation and an increase in its accumulation in the brain of mice with IDE knockout [144]. At the same time, a decrease in IDE expression and its catalytic activity was observed in patients with a familial form of AD [145,146].

As mentioned above, the development of T2DM is accompanied by IR [103]. It leads to an increase in ROS production and the initiation of the formation of the NLPR3 inflammasome, as well as obesity. It also increases the production of proinflammatory cytokines by astroglia and microglia, which might end in neuronal degeneration [137,147]. Thus, elderly people with obesity, T2DM, or hypercholesterolemia are more susceptible to AD [148]. Besides, this neurodegenerative disease is considered by some authors as «type 3 diabetes mellitus» [149,150].

### 4.4. Major Depressive Disorder and Alzheimer’s Disease

According to the WHO Guidelines on risk reduction of cognitive decline and dementia, depression is one of the main risk factors for dementia [22].

In recent years, the number of studies devoted to the meta-analysis of the relationship between depressive disorders, including major depressive disorder (MDD), and AD has increased [151,152,153]. It was shown that the risk of dementia, including Alzheimer’s type, increases by 14% even with a single depressive episode per lifetime [154]. Additionally, the duration and clinical severity of each depressive episode correlate with the incidence of neurodegenerative diseases [155]. Therapy of patients with depressive disorders with selective serotonin reuptake inhibitors (SSRIs) led to a decrease in the risk of developing dementia, but still, this indicator remained higher than the average in the population [28].

In depression, patients experience a proinflammatory background similar to inflammaging, which leads to the activation of microglia, the death of astrocytes, and somatostatin-positive interneurons [28]. According to a meta-analysis by Köhler et al. [156], there is a significant increase in the content of proinflammatory (IL-6, TNF, sIL-2, CCL2, IL-13, IL-18, IL-12) and anti-inflammatory cytokines (IL-10, IL-1RA, and sTNFR 2 (soluble TNF receptor 2)) in the blood serum of patients with MDD, compared with a group of healthy volunteers [157]. Cheng et al. [157], by using a model of stress learned helplessness with depression-like behavior, found that TLR4−/− C57Bl/6 mice have reduced production of proinflammatory (TNFα, IL-6, IFNκ, IL-17A) and immunomodulatory cytokines (IL-12(p70), IL-2, IL-3, IL-5, IL-10, IL-13, IL-12) [p40], and chemokines (CXCL1, CCL11, CCL2, CCL3, CCL4, CCL5) [158] compared to wild-type mice. The authors showed that stress exposure caused an increase in the expression of cytokines in the hippocampus but not in the prefrontal cortex in experimental animals [158]. This may be due to a denser population of the hippocampus with microglia during antenatal development [159]. It has also been reported that patients with MDD monocytes showed an overexpression of the genes forming an inter-correlating gene cluster (cluster 3), including a subcluster of mitochondrial apoptosis/growth regulating genes (BAX, BCL10, EGR1, and EGR2), various genes previously described in athero-protective M(hb) macrophages (ABCA1, ABCG1, NR1H3, MRC1, CD163), the gene for pro-apoptotic/pro-inflammatory TNF, the gene for the immune regulating/protein chaperone molecule HSP70, and the cholesterol pathway gene MVK. This phenotype provides disturbances of monocyte functioning, such as mitochondrial apoptosis, the unfolded protein response and the cholesterol shuttle. All these features are typical for cell senescence, which lead to SASP producing that in turn is a main characteristic of inflammaging [160].

One of the mechanisms for the development of AD against the background of MDD may be the activation of the hypothalamic-pituitary-adrenal axis with an increase in the synthesis of glucocorticoids (GCs) [28]. It was established that the number of receptors for GCs on the neurons decreases in the hypothalamus and pituitary gland due to an increase in GCs in the blood and a violation of the mechanism of reverse negative regulation of their expression during the development of depression [161]. In turn, GCs activate APP expression and tau protein phosphorylation in the brain, in particular, in the hippocampus [162].

Green et al. found an increase in the level of Aβ by 60% in the brain 3xTg-AD mice after administration of dexamethasone, compared with the control group. Also, an increase in the number of C99 fragments of APP was observed, which indirectly indicates an increase in BACE1 activity and the accumulation of total, but not phosphorylated tau protein [163]. In a model of depression in rats (chronic unpredictable mild stress-induced depression) by Western blotting, an increase in the level of phosphorylated tau protein was revealed in the hippocampus and prefrontal cortex of the experimental group compared with the group of animals that received the SSRI fluoxetine [164]. The anti-inflammatory effects of SSRIs and tricyclic antidepressants are likely mediated through 5-HTT receptors, increased intracellular cAMP, increased expression of IL-10 by microglia, and the inhibition of the NF-κB signaling pathway [165].

To sum up, the development of AD is often associated with chronic inflammatory diseases, including atherosclerosis, obesity/metabolic syndrome, T2DM, and major depressive disorder (Figure 1). The progression of these diseases is based on the development of an inflammatory immune response, which can be activated by various signaling pathways, including NF-κB, NLRP3, IRAK1, IKK, etc. Activation of systemic inflammatory reactions may lead to (1) an increase in the permeability of the BBB and the hemato-liquor barrier, (2) polarization of microglia and astroglia according to the proinflammatory M1 and A1 phenotype, (3) triggering of the amyloid pathway of APP metabolism, and (4) hyperphosphorylation of tau protein with the formation of amyloid plaques and NFTs. All these processes contribute to the loss of synapses and neurodegeneration. The likelihood of developing chronic inflammatory diseases increases with age, possibly due to cellular aging and increased SASP production, which are hallmarks of inflammaging.

## 5. Inflammaging and Alzheimer’s Disease

Inflammation plays a crucial role in the pathogenesis of various neurological disorders and in mediating interactions between the immune and nervous systems, which may be the key to preventing or delaying the onset of most CNS diseases. Local inflammation in the CNS is observed in amyotrophic lateral sclerosis, multiple sclerosis, Parkinson’s disease, and AD [166]. Among the potential mechanisms contributing to chronic inflammation in the CNS in AD, the neuroinflammation hypothesis is dominant. In accordance with this, DAMPs formed during cell damage activate TLR4 on glial cells, which induces inflammatory reactions [167]. Activation of TLRs leads to the assembly of NLRP3, which plays a pivotal role in chronic inflammation in obesity, IR, T2DM, and neurodegenerative diseases. An increase in the level of muramyl dipeptide, which is a component of the bacterial cell wall, was detected in the blood of old mice [168], and a compensatory increase in miR-223 siRNA [169], which inhibits NLRP3 translation [170] in response to developing chronic inflammation, was also observed [171].

DAMPs are formed in the inflammatory process due to caspase-dependent necrotic cell death, known as necroptosis, which may be one of the mechanisms of inflammaging [172]. Royce et al. [171] showed that the level of markers of necroptosis, which is phosphorylated and non-phosphorylated mixed lineage kinase domain, like pseudokinase (MLKL) in white adipose tissue in mice, increases 3-fold with age. Age-related activation of necroptosis markers was accompanied by an increase in the production of IL-6, TNF-α, and IL-1β and several chemokines [171]. According to Caccamo et al. [173], an increase in the expression of necroptosis markers was observed in the brain of AD patients, and the activation of necroptosis in APP/PS1 transgenic mice with AD increased cognitive impairment. Inhibition of necroptosis by necrostatin-1 (a blocker of TNF-1α-dependent necrosis) in APP/PS1 mice resulted in a decrease in the number of amyloid plaques, the production of proinflammatory cytokines TNF-α and IL-1β, and an improvement in cognitive functions [172]. This indicates an association between age-related activation of necroptosis and the development of neurodegenerative diseases, including AD.

One of the processes that also determines the development of inflammaging is autophagy/mitophagy, the destruction of damaged organelles/mitochondria and protein aggregates by the cell, which is impaired with age [101,174]. Accumulations of damaged organelles in cells as a result of inadequate autophagy, as well as mitochondrial DNA formed by mitochondria not subjected to mitophagy, are considered DAMPs and are recognized by PRRs of immunocompetent cells [175,176]. The interaction of DAMPs with PRRs activates the transcription of IL-6, TNF-α, IL-1β, MMP-8, and NLRP3, being key regulators of inflammation [177,178], the increase of which leads to caspase activation and pyroptosis [179].

According to the literature, microglia activation according to the proinflammatory M1 phenotype is observed with aging. It combines with an increase in the production of proinflammatory cytokines by these cells, including IL-1β, TNF-α, and IL-6 [180]. Microglial cells activated via the M1 pathway are found around amyloid plaques and neurons containing neurofibrils in AD [181]. Microglia recognizes Aβ through a number of receptor complexes, including CD14, TLR2, TLR4, α6β1 integrin, CD47, and CD36, and are capable of phagocytizing amyloid [182]. On the one hand, the accumulation of Aβ in the brain of AD patients may be associated with impaired phagocytosis of amyloid by microglia [183]. On the other hand, as a result of inflammaging, prolonged activation of microglia by a proinflammatory phenotype can initiate the formation of Aβ along the amyloid pathway and neurodegeneration [184]. Activation of microglia in AD is combined with mitochondrial dysfunction, decreased oxidative phosphorylation, and the activation of glycolysis [185]. However, despite the huge number of works devoted to the study of the pathogenesis of AD, it has not yet been possible to establish whether glia-associated inflammation in AD is a cause or a consequence of neurodegeneration.

Thus, processes such as cellular aging, leading to senescence of the immune and nervous systems, mitochondrial dysfunction, SASP production, activation of auto- and mitophagy, development of hypoxia, and the activation of HIF1-dependent signaling pathways all play an important role in the development of inflammaging (Figure 2).

## 6. The Role of Adjuvant Therapy in AD Development

It has been proposed that anti-inflammatory compounds have the potential to treat those suffering from AD and other neurodegenerative diseases. Some researchers suggest a protective effect of non-selective nonsteroidal anti-inflammatory drugs (NSAIDs) against neurodegeneration [186,187].

NSAIDs are considered blockers of cyclooxygenase (COX), which leads to a decrease in prostaglandins, prostacyclin, and thromboxane levels. Also, as a modulator of APP processing, NSAID might probably play a role in inhibiting the Aβ formation and reduction of amyloidogenic forms of APP [188]. A meta-analysis including 16 investigations demonstrates that present or previous utilization of NSAIDs is linked to a decreased relative risk of AD [189]. Still, other studies did not find this association. Additional longstanding, controlled studies examining anti-inflammatory drugs, including tarenflurbil in mild AD patients [189] and prednisone [190] and celecoxib [191] in mild-to-moderate AD patients, report the presence of detrimental consequences vs. placebo [192]. This fact may be due to the side effects of these drugs. NSAIDs might be useful for AD prevention when their administration occurs years before the usual onset age; however, when used later in life, they might increase the risk of disease or accelerate its progression [192].

Insulin is suggested to have neuroprotective properties and exert neurotrophic effects on CNS neurons [193,194]. In patients with mild cognitive impairment and late-onset AD, intranasal insulin administration improves cerebral glucose metabolism and preserves the volume of brain regions affected by AD pathology [195].

Metformin reduces insulin-mediated hepatic glucose production, increases insulin sensitivity, and has a neuroprotective effect on human neural stem cells, restoring mitochondrial functions and attenuating age effects [196,197]. This is the main first-line medication for the treatment of T2DM, particularly in overweight people [198,199]. It was shown that mild cognitive impairment [200,201] and dementia [202,203] were less pronounced in persons with diabetes taking metformin compared with others with no medication or taking other glucose-lowering agents. Other drugs used to treat diabetes also reduce the severity of neurodegenerative changes [194]. Thiazolidinediones (TZDs), pioglitazone, and rosiglitazone showed neuroprotective effects in AD due to the inhibition of inflammatory gene expression and the alteration of amyloid beta generation and deposition [204]. Pioglitazone may provide an improvement in the early stages and in mild-to-moderate AD [205,206]. Glucagon-Like Peptide-1 (GLP-1) Receptor Agonists also have a neuroprotective role in the brain of AD mouse models [207]. The positive effects of drugs used to treat diabetes once again confirm the multifactorial nature of AD.

## 7. Conclusions

Aging is an important risk factor for the development of age-related diseases such as atherosclerosis, T2DM, and neurodegenerative diseases, including AD. Activation of cells of the proinflammatory phenotype is observed with aging, producing a wide range of proinflammatory factors SASP, which leads to the development of inflammaging.

The mechanisms of AD are not well understood. The main theory for AD development is the “amyloid cascade” hypothesis. However, the initial mechanisms for forming insoluble forms of amyloid and NFTs are still unclear.

The formation of senile plaques and NFTs is more often observed in individuals with chronic inflammatory diseases, including atherosclerosis, obesity/metabolic syndrome, T2DM, and major depressive disorder. The pathogenesis of these diseases is based on developing an inflammatory immune response, which can be activated in various ways, including through NF-κB, the NLRP3 inflammasome, several IRAK1, IKK kinases, etc. Cellular aging, mitochondrial dysfunction, SASP production, activation of auto- and mitophagy, and the development of hypoxia play a pivotal role in the development of inflammaging.

However, despite all people developing inflammaging with age, not everyone suffers from age-related neurodegenerative diseases, such as AD. That means chronic inflammation correlates with the risk of developing age-related neurodegenerative diseases, but for AD development, the inflammatory process alone is not enough. This confirms the multiplicity of mechanisms that activate the processes of neurodegeneration in AD.

Thus, further studies of inflammaging, taking into account the individual characteristics of its course and comorbidity of patients, will make it possible to establish the initial mechanisms for the development of AD and to identify new approaches to the treatment of this disease.

## Figures and Tables

**Figure 1 brainsci-12-01237-f001:**
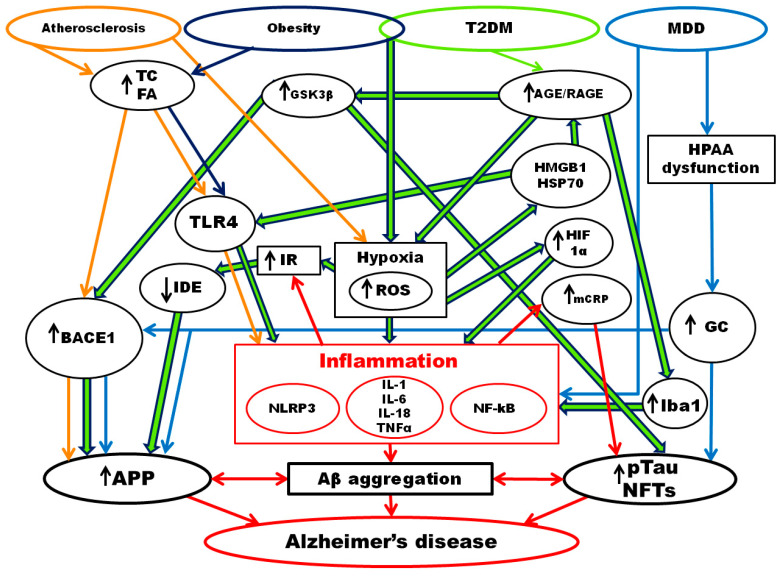
Molecular mechanisms of the relationship between chronic inflammatory diseases and AD. T2DM—type 2 diabetes mellitus; MDD—major depressive disorder; TC—total cholesterol; FA—fatty acids; HPAA—hypothalamic-pituitary-adrenal axis; TLR-4—toll-like receptors type 4; AGE/RAGE—advanced products of glycation/ advanced products of glycation receptors; BACE1—beta-site amyloid precursor protein cleaving enzyme 1, β-secretase; GSK3β—Glycogen synthase kinase 3β; ROS—reactive oxygen species; IR—insulin resistance; IDE—insulin-degrading enzyme; HMGB1—high mobility group box 1; HSP70—heat shock protein 70; HIF1α—Hypoxia-inducible factor 1-alpha; GC—glucocorticoids; Iba1—Ionized calcium-binding adaptor molecule 1; mCRP—monomeric C-reactive protein; NLRP3—NLR family pyrin domain containing 3; TNFα—tumor necrosis factor α; NF-κB—Nuclear factor kappa B; APP—Amyloid precursor protein; pTau—phosphorylated tau protein; NFTs—neurofibrillary tangles.

**Figure 2 brainsci-12-01237-f002:**
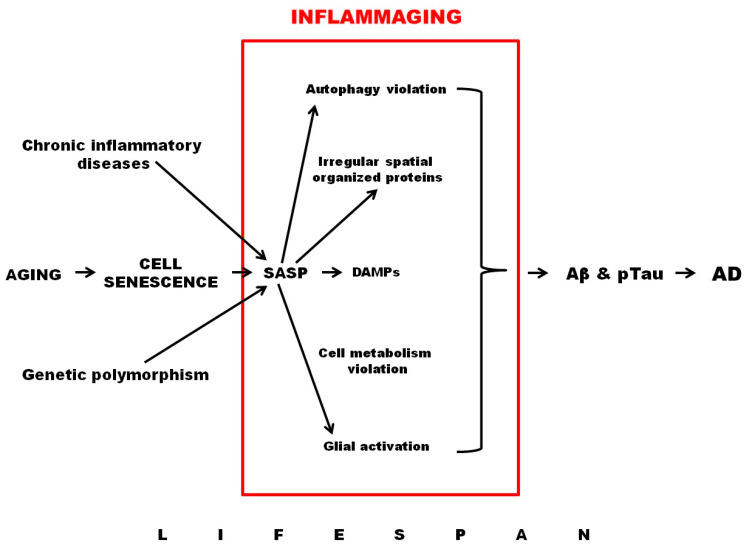
Role of inflammaging in AD development.

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
