# Peer review of "Alzheimer’s Disease and Inflammaging"

_brainsci, 2022, doi:10.3390/brainsci12091237_

Round 1

Reviewer 1 Report

The largest concern about this manuscript was the poor English grammar and errors that make it difficult to read and comprehend. The grammatical mistakes are too numerous to list individually but the authors should have a native English speaker provide editing assistance. The second major concern is the lack of focus and organization of the manuscript with excessive detail given in many areas of the manuscript which are not central to the theme of inflammaging as a risk factor for AD, for example the degree of detail given to the cleaving of amyloid protein lines 179-187 is not central to the manuscript and distracting to the reader; there are many other instances of similar departures from the theme of inflammaging as an AD risk factor and major edits for organization are needed to ensure that all paragraphs under each heading are related to the heading (for example lines 256-269 are totally unrelated to the heading entitled “Theories of the development of Alzheimer’s disease”).

Although this review highlighted the potential cellular and biochemical mechanisms of possible inflammaging and much of the evidence was derived from mouse studies, it was limited in scope in terms of review of human and clinical studies which potential bearing on inflammaging including studies of DM2 in AD, obesity in AD as well as evidence as to the effect size of each of this conditions as AD risk factors.

Suggest also referring to:

https://www.nature.com/articles/nrneurol.2010.130

 https://doi.org/10.1186/1742-2094-5-51

There was no discussion as to modulating factors that combat inflammaging and whether treatments of inflammation or underlying diseases and conditions implicated: DM2, obesity, atherosclerosis can decrease risk of AD. What about other inflammatory conditions including primary infectious processes, autoimmune disorders, do they increase AD risk and by the same mechanism? What about people treated with NSAIDs, steroids and other inflammatory conditions and how do these factors modulate AD risk? What is the state of knowledge regarding the effect if any of cerebrovascular disease due to DM2, HLD and other conditions implicated here as an additional mediating factor in dementia due to non-AD related pathology that could be contributing to clinical picture of vascular risk factors worsening AD?

It’s not clear why depression was included in this review as its not typically a disease of aging and although it may involve inflammatory components, is not traditionally considered a disorder of inflammation. If depression is included here it leads readers to question why other brain disorders that are non-age dependent and may have more direct inflammatory mechanisms were not included here including Multiple Sclerosis and other inflammatory disorders. It is this reviewers opinion that depression should not be included in the review to best focus scope on inflammatory disorders more associated with aging.

The logic discussed is unclear as to whether systemic inflammation associated with aging independently increases AD risk directly or whether inflammation associated with aging is an AD risk factor which is dependent upon other disease processes or conditions including DM2, obesity, atherosclerosis.

There is little contrary or opposing evidence presented as to the view that DM2, obesity, hypercholesterolemia and depression all increase AD risk through inflammation. One example of an opposing line of evidence was given in lines 300-302 but presenting more opposing evidence would reduce any impression of bias. Also proposing explanatory reasons as to why studies and lines of evidence did not agree would strengthen this review. This could be done following lines 306-308 for example and in other places.

Author Response

Dear reviewer, we are very thankful for your huge work under our manuscript. We had tried to improve our article according to your remarks and questions. We tried to answer all your questions in the text and added citations according to your remarks in the notes, included in the body of the manuscript.

  1. The largest concern about this manuscript was the poor English grammar and errors that make it difficult to read and comprehend. The grammatical mistakes are too numerous to list individually but the authors should have a native English speaker provide editing assistance.

We turned to a native English-speaking person for help and made corrections to the manuscript.

  1. The second major concern is the lack of focus and organization of the manuscript with excessive detail given in many areas of the manuscript which are not central to the theme of inflammaging as a risk factor for AD, for example the degree of detail given to the cleaving of amyloid protein lines 179-187 is not central to the manuscript and distracting to the reader; there are many other instances of similar departures from the theme of inflammaging as an AD risk factor and major edits for organization are needed to ensure that all paragraphs under each heading are related to the heading (for example lines 256-269 are totally unrelated to the heading entitled “Theories of the development of Alzheimer’s disease”).

We tried to improve section 3 «Hypothesis of the development of Alzheimer's disease»  and reduced the fragment about amyloid cascade hypothesis and BBB permeability.

  1. Although this review highlighted the potential cellular and biochemical mechanisms of possible inflammaging and much of the evidence was derived from mouse studies, it was limited in scope in terms of review of human and clinical studies which potential bearing on inflammaging including studies of DM2 in AD, obesity in AD as well as evidence as to the effect size of each of this conditions as AD risk factors. Suggest also referring to:

https://www.nature.com/articles/nrneurol.2010.130

 https://doi.org/10.1186/1742-2094-5-51

 In the sections on the relationship of AD with obesity (lines 379-381) and DM2 (lines 406-409), we cited articles that show relationship between increase of AD development and these diseases. We did not find any other relevant studies.

  1. There was no discussion as to modulating factors that combat inflammaging and whether treatments of inflammation or underlying diseases and conditions implicated: DM2, obesity, atherosclerosis can decrease risk of AD.

We have added a fragment about the use of NSAIDs drugs and drugs using for DM2 treatment and their association with AD together with information concerning the therapy of other discussed diseases and its impact on AD processing (section 6).

  1. What about other inflammatory conditions including primary infectious processes, autoimmune disorders, do they increase AD risk and by the same mechanism?

We added this information to the manuscript in section 3, subsections Inflammation and AD development and Autoimmune hypothesis of AD development.

  1. What about people treated with NSAIDs, steroids and other inflammatory conditions and how do these factors modulate AD risk?

We have added a fragment about the use of NSAIDs drugs and their association with AD together with information concerning the therapy of other discussed diseases and its impact on AD processing (section 6).

  1. What is the state of knowledge regarding the effect if any of cerebrovascular disease due to DM2, HLD and other conditions implicated here as an additional mediating factor in dementia due to non-AD related pathology that could be contributing to clinical picture of vascular risk factors worsening AD?

We have tried to consider cerebrovascular disease as risk factor of AD (lines 428-437).

Сerebrovascular changes such as hemorrhagic infarcts, small and large ischemic cortical infarcts, vasculopathies, and changes in cerebral white matter are known to increase the risk of dementia, including AD (Silva MVF, Loures CMG, Alves LCV, de Souza LC, Borges KBG, Carvalho MDG. Alzheimer's disease: risk factors and potentially protective measures. J Biomed Sci. 2019 May 9;26(1):33. doi: 10.1186/s12929-019-0524-y.). In a longitudinal community-based clinical-pathologic cohort study, 38% of persons with dementia had AD and infarcts, and 30% of them had pure AD (Schneider JA, Arvanitakis Z, Bang W, Bennett DA. Mixed brain pathologies account for most dementia cases in community-dwelling older persons. Neurology 2007; 69: 2197–2204).

In a large cohort of women diagnosed with breast cancer at age≥65 with up to 26 years of follow-up from 1991 to 2016 it was found that cerebrovascular disease (CVD), stroke, and diabetes are associated with a significantly higher risk of AD, whereas hypertension was associated with a significantly lower risk of AD in those aged≥75 years (Du XL, Song L, Schulz PE, Xu H, Chan W. Risk of Developing Alzheimer's Disease and Related Dementias in Association with Cardiovascular Disease, Stroke, Hypertension, and Diabetes in a Large Cohort of Women with Breast Cancer and with up to 26 Years of Follow-Up. J Alzheimers Dis. 2022;87(1):415-432. doi: 10.3233/JAD-215657). 

In patients with diabetes, arteriosclerosis, and cellulosic necrosis and amyloidosis there are pathological changes of blood vessels with destruction of BBB, cerebral endothelial cell dysfunction, reduced cerebral blood flow thus eventually leading to brain parenchymal damage (Liang Z, Wu L, Gong S, Liu X. The cognitive dysfunction related to Alzheimer disease or cerebral small vessel disease: What's the differences. Medicine (Baltimore). 2021 Aug 27;100(34):e26967. doi: 10.1097/MD.0000000000026967.). According to (Liang Z, Wu L, Gong S, Liu X. The cognitive dysfunction related to Alzheimer disease or cerebral small vessel disease: What's the differences. Medicine (Baltimore). 2021 Aug 27;100(34):e26967. doi: 10.1097/MD.0000000000026967.) diabetes is the risk factor both for AD and sporadic cerebral small vessel disease (CSVD). It is known that with age large artery stiffness increases that also could lead to blood vessels dysfunction, resulting in neuropathology, and cognitive impairment (Kehmeier MN, Walker AE.Sex Differences in Large Artery Stiffness: Implications for Cerebrovascular Dysfunction and Alzheimer's Disease. Front Aging. 2021 Dec;2:791208. doi: 10.3389/fragi.2021.791208.). 

However, the cognitive dysfunctions of AD are difference to that of sporadic cerebral small vessel disease (CSVD). According to Liang Z, et al. (Liang Z, Wu L, Gong S, Liu X. The cognitive dysfunction related to Alzheimer disease or cerebral small vessel disease: What's the differences. Medicine (Baltimore). 2021 Aug 27;100(34):e26967. doi: 10.1097/MD.0000000000026967.) compared with AD, CSVD has severe impairments in the area of immediate memory, attention and calculation ability, language fluency, abstract thinking and execution ability, but the orientation ability of CSVD patient is better than that of AD patient.

It is possible that another additional mechanism in the development of AD is a violation of the cerebral microvasculature, however, this factor is not decisive and often cerebrovascular diseases are not accompanied by the development of AD.

  1. It’s not clear why depression was included in this review as its not typically a disease of aging and although it may involve inflammatory components, is not traditionally considered a disorder of inflammation. If depression is included here it leads readers to question why other brain disorders that are non-age dependent and may have more direct inflammatory mechanisms were not included here including Multiple Sclerosis and other inflammatory disorders. It is this reviewers opinion that depression should not be included in the review to best focus scope on inflammatory disorders more associated with aging.

According to the WHO Guidelines on risk reduction of cognitive decline and dementia depression is one of main risk factors of dementia unlike such diseases as multiple sclerosis or some autoimmune disorders (WHO. Risk reduction of cognitive decline and dementia: WHO guidelines. 2019. Available at: https://www.who.int/publications/i/item/9789241550543 accessed on 10 August 2022). Also it has been reported that in patients with MDD monocytes showed an overexpression of the genes forming an inter-correlating gene cluster (cluster 3) including subcluster of mitochondrial apoptosis/growth regulating genes (BAX, BCL10, EGR1, and EGR2), various genes previously described in M(hb) macrophages (ABCA1, ABCG1, NR1H3, MRC1, CD163), the gene for pro-apoptotic/pro-inflammatory TNF, the gene for the immune regulating/protein chaperone molecule HSP70, and the cholesterol pathway gene MVK. This phenotype provides disturbances of monocyte functioning, such as mitochondrial apoptosis, the unfolded protein response, and the cholesterol shuttle. All these features are typical for cell senescence, which leading to SASP producing that in turn is a main characteristic of inflammaging (Simon, M.S.; Schiweck, C.; Arteaga-Henríquez, G. et al. Monocyte mitochondrial dysfunction, inflammaging, and inflammatory pyroptosis in major depression. Prog Neuropsychopharmacol Biol Psychiatry,  2021, 111, 110391). Hence, it could be considered as a condition enhancing the inflammation level in patients, which might influence on worsening of AD progression.

  1. The logic discussed is unclear as to whether systemic inflammation associated with aging independently increases AD risk directly or whether inflammation associated with aging is an AD risk factor which is dependent upon other disease processes or conditions including DM2, obesity, atherosclerosis.

We have edited the conclusions section to clarify the role of inflammation in AD development (lines 616-620).

  1. There is little contrary or opposing evidence presented as to the view that DM2, obesity, hypercholesterolemia and depression all increase AD risk through inflammation. One example of an opposing line of evidence was given in lines 300-302 but presenting more opposing evidence would reduce any impression of bias. Also proposing explanatory reasons as to why studies and lines of evidence did not agree would strengthen this review. This could be done following lines 306-308 for example and in other places.

We added a fragment concerning probable impact of T2DM on AD progression via vascular disturbances (lines 431-436).

Reviewer 2 Report

Dear Authors,

 I thank You for the chance to read this manuscript, submitted for publication in Brain Sciences.

Here are my comments and suggestions. I hope that they are useful.

MAJOR COMMENTS

You should address the role of inflammation also in patients with early-onset Alzheimer ‘s disease (AD), highlighting differences (if any) with the type of inflammation present in patients with elderly-onset AD. What You wrote in lines 157-162 is not sufficient.      

The role of anti-inflammatory drugs as proposed way of treating patients with AD should be at least mentioned.

Finally, what You wrote in lines 556-562 seemed contradictory. Indeed, elderly-onset AD is more likely to develop in people with chronic diseases, and inflammaging is only one connecting factor.  Please, modify these phrases in the newer version of Your manuscript.   

MINOR COMMENTS  

Please, write Abbreviations before References and not at the beginning of your manuscript.

Grammar needs to be improved, and punctuation should be revised. Please, consider to review the whole text with a better English editing.  As an example, paragraph no. 2 is unclear because repetitive phrases are present. My suggestion is to simplify concepts by writing more linear sentences, and to remove the repetitive phrases (such as the lines from 136 to 141). As another example, the punctuation should be revised in Abstract; and so on….

Discussion: consider a figure that summarizes the role of inflammaging in AD. It can be useful to readers.

Author Response

Dear reviewer, we are thankful to you for your work under our manuscript. We are very grateful to you for your comments and remarks. Without any doubt, they helped to make our work better and made it possible to pay attention to the shortcomings. We tried to take them into consideration in case to rework some mentioned points and hope that this will make it possible to publish our article.

MAJOR COMMENTS

  1. You should address the role of inflammation also in patients with early-onset Alzheimer‘s disease (AD), highlighting differences (if any) with the type of inflammation present in patients with elderly-onset AD. What You wrote in lines 157-162 is not sufficient.      

We have added a fragment concerning polymorphisms of the IL-1RA gene and the risks of developing AD of early onset (lines 138-144).

  1. The role of anti-inflammatory drugs as proposed way of treating patients with AD should be at least mentioned.

We have added a section about the use of NSAIDs drugs and their association with AD (section 6).

  1. Finally, what You wrote in lines 556-562 seemed contradictory. Indeed, elderly-onset AD is more likely to develop in people with chronic diseases, and inflammaging is only one connecting factor.  Please, modify these phrases in the newer version of Your manuscript.   

We have tried to modify these lines. We agree with your opinion; this idea was implied.

MINOR COMMENTS  

  1. Please, write Abbreviations before References and not at the beginning of your manuscript.

We replaced Abbreviations before References.

  1. Grammar needs to be improved, and punctuation should be revised. Please, consider to review the whole text with a better English editing.  As an example, paragraph no. 2 is unclear because repetitive phrases are present. My suggestion is to simplify concepts by writing more linear sentences, and to remove the repetitive phrases (such as the lines from 136 to 141). As another example, the punctuation should be revised in Abstract; and so on….

We turned to a native English-speaking person for help and made corrections to the manuscript.

  1. Discussion: consider a figure that summarizes the role of inflammaging in AD. It can be useful to readers.

We tried to create one more figure (Fig.2 section 5).

Reviewer 3 Report

Dear Authors, 

The manuscript titled "ALZHEIMER’S DISEASE AND INFLAMMAGING" is well written and organized.

However, I have some suggestions and comments in order to improve the quality and the update of your paper.

1- Concerning the term inflammaging, it should be fine to mention the first article focusing on this new term (doi: 10.1111/j.1749-6632.2000.tb06651.x).

2- In the point 3, concerning the "Theories of the development of Alzheimer's disease", it should be fine to mention the loss of cholinergic neurons in a particolar region of interest that is the nucleus basalis of Meynert. On of the first paper mentioning this theory come back to 1983 (doi: 10.1007/BF00697388). However, still now the nucleus basalis of Meynert is of interest for alternative therapies (10.1038/mp.2014.32) but also to explain not only AD but also other neurological pathologies (DOI:10.1017/S1041610220003944). 

3- Focusing on the inflammatory cytokines, recently has been demonstrated that TNFα double affect the human cholinergic neurons from nucleus basalis of Meynert both inducing differentiation and decreasing neurogenesis, also by epigenetic mechanisms (doi: 10.3390/ijms21176128).

5- Finally, a mention should be done on gender differences. Indeed, in this regard, has been hypothesized the role of estrogens (doi: 10.3389/fnagi.2019.00315) and the protective role of phytoestrogens against the neuroinflammation (doi: 10.1002/biof.1701).

6- Few typos are present in the main text (use β symbols and not "beta"; use the line "amyloid-β" or not "amyloid β"; "amiloid-β" or "β-amyloid"; page 4, line 95 "inflammaging" is written in bold and italics for a half). Please, read again carefully the manuscript in order to avoid typos.

7- The idea of a picture like this is fine. However, in my opinion is to rough. Moreover, there is a double arrow in the IR (insuline resistence) circle. Please, edit the figure.

Author Response

Dear Reviewer! We would like to thank you for careful evaluation of our manuscript and for the valuable comments. Please find below the detailed answers to your remarks.

  • Concerning the term inflammaging, it should be fine to mention the first article focusing on this new term (doi: 10.1111/j.1749-6632.2000.tb06651.x).

We cited this article you mentioned.

  • In the point 3, concerning the "Theories of the development of Alzheimer's disease", it should be fine to mention the loss of cholinergic neurons in a particular region of interest that is the nucleus basalis of Meynert. One of the first paper mentioning this theory come back to 1983 (doi: 10.1007/BF00697388). However, still now the nucleus basalis of Meynert is of interest for alternative therapies (10.1038/mp.2014.32) but also to explain not only AD but also other neurological pathologies (DOI:10.1017/S1041610220003944). 

We added the fragment concerning the nucleus basalis of Meynert (lines 185-193).

  • Focusing on the inflammatory cytokines, recently has been demonstrated that TNFα double affect the human cholinergic neurons from nucleus basalis of Meynert both inducing differentiation and decreasing neurogenesis, also by epigenetic mechanisms (doi: 10.3390/ijms21176128).

We added the fragment concerning the TNF effects on differentiation and decreasing neurogenesis (lines 190-193).

  • Finally, a mention should be done on gender differences. Indeed, in this regard, has been hypothesized the role of estrogens (doi: 10.3389/fnagi.2019.00315) and the protective role of phytoestrogens against the neuroinflammation (doi: 10.1002/biof.1701).

We added the fragment concerning estrogen’s influence on AD development in section 3, subsection Oestrogen hypothesis of AD development.

6- Few typos are present in the main text (use β symbols and not "beta"; use the line "amyloid-β" or not "amyloid β"; "amiloid-β" or "β-amyloid"; page 4, line 95 "inflammaging" is written in bold and italics for a half). Please, read again carefully the manuscript in order to avoid typos.

We’ve edited the text according to your remarks.

7- The idea of a picture like this is fine. However, in my opinion is to rough. Moreover, there is a double arrow in the IR (insuline resistence) circle. Please, edit the figure.

We’ve added the fixed figure.

Round 2

Reviewer 2 Report

Dear Authors,

all my comments and suggestions were satisfactorily met in the revised version of your manuscript.